# Scoring Workers in Crowdsourcing: How Many Control Questions are Enough?

**Qiang Liu**
Dept. of Computer Science
Univ. of California, Irvine
qliu1@uci.edu

**Mark Steyvers**
Dept. of Cognitive Sciences
Univ. of California, Irvine
mark.steyvers@uci.edu

**Alexander Ihler**
Dept. of Computer Science
Univ. of California, Irvine
ihler@ics.uci.edu

## Abstract

We study the problem of estimating continuous quantities, such as prices, probabilities, and point spreads, using a crowdsourcing approach. A challenging aspect of combining the crowd's answers is that workers' reliabilities and biases are usually unknown and highly diverse. Control items with known answers can be used to evaluate workers' performance, and hence improve the combined results on the target items with unknown answers. This raises the problem of how many control items to use when the total number of items each workers can answer is limited: more control items evaluates the workers better, but leaves fewer resources for the target items that are of direct interest, and vice versa. We give theoretical results for this problem under different scenarios, and provide a simple rule of thumb for crowdsourcing practitioners. As a byproduct, we also provide theoretical analysis of the accuracy of different consensus methods.

## 1 Introduction

The recent rise of crowdsourcing has provided a fast and inexpensive way to collect human knowledge and intelligence, as illustrated by human intelligence marketplaces such as Amazon Mechanical Turk, games with purpose like ESP, reCAPTCHA, and crowd-based forecasting for politics and sports. One of the philosophies behind these successes is the *wisdom of crowds* phenomenon: properly combining a group of untrained people can be better than the average performance of the individuals, and even as good as the experts in many application domains (e.g., Surowiecki, 2005, Sheng et al., 2008). Unfortunately, it is not always obvious how best to combine the crowd, because the (often anonymous) workers have unknown and diverse levels of expertise, and potentially systematic biases across the crowd. Naïve consensus methods which simply take uniform averages or the majority answer of the workers have been known to perform poorly. This raises the problem of *scoring the workers*, that is, estimating their expertise, bias and any other associated parameters, in order to combine their answers more effectively.

One direct way to address this problem is to score workers using their past performance on similar problems. However, this is not always practical, since historical records are hard to maintain for anonymous workers, and their past tasks may be very different from the current one. An alternative is the idea behind reCAPTCHA: "seed" some control items with known answers into the assigned tasks (without telling workers which are control items), score the workers using these control items, and weight their answers accordingly on the unknown target items. Similar ideas have been widely used in existing crowdsourcing systems. CrowdFlower, for example, provides interfaces and tools to allow requesters to explicitly specify and analyze a set of control items (sometimes called "gold data"). The reCAPTCHA example is a particularly simple case, where workers answer exactly one control and one target item. However, in general crowdsourcing, the workers may answer hundreds of questions, raising the question of how many control items should be used. There is a clear trade-off: having workers answer more control items gives better estimates of their performance and any

potential systematic bias, but leaves fewer resources for the target items that are of direct interest. However, using few control items gives poor estimates of workers' performance and their biases, also leading to bad results. A deep understanding of the *value* of control items may provide useful guidance for crowdsourcing practitioners.

On the other hand, a line of research has studied more advanced consensus methods that are able to simultaneously estimate the workers' performance and items' answers without any ground truth on the items, by building joint statistical models of the workers and labels (e.g., Dawid and Skene, 1979, Whitehill et al., 2009, Karger et al., 2011, Liu et al., 2012, Zhou et al., 2012). The basic idea is to score the workers by their agreement with other workers, assuming that the majority of workers are correct. Perhaps surprisingly, the worker reliabilities estimated by these "unsupervised" consensus methods can be almost as good as those estimated when the true labels of all the items are known, and are much better than self-evaluated worker reliability (Romney et al., 1987, Lee et al., 2012). Control items can also be incorporated into these methods: but how much can we expect them to improve results, or does an "unsupervised" method suffice?

The goal of this paper is to study the value of control items, and provide practical guidance on how many control items are enough under different scenarios. We give both theoretical and empirical results for this problem, and provide some rules of thumbs that that are easy to use in practice. We develop our theory on a class of Gaussian models for estimating continuous quantities, such as forecasting probabilities and point spreads in sports games, and show how it extends to more general models. As a byproduct, we also provide analytic results of the accuracy of different consensus algorithms. Important practical issues such as the impact of model misspecification, systematic biases and heteroscedasticity are also highlighted on real datasets.

## 2   Background

Assume there is a set $\mathcal{T}$ of *target* items, associated with a set of labels $\mu_{\mathcal{T}} := \{\mu_i : i \in \mathcal{T}\}$ whose true values $\mu_{\mathcal{T}}^*$ we want to estimate. In addition, we have a set $\mathcal{C}$ of *control* (or *training*) items whose true labels $\mu_{\mathcal{C}}^* := \{\mu_i^* : i \in \mathcal{C}\}$ are known. We denote the set of workers by $\mathcal{W}$; each worker $j$ is associated with a parameter $\nu_j^*$ that characterizes their expertise, bias, any other relevant features. We denote the complete vector of worker parameters by $\nu := \{\nu_j^* : j \in \mathcal{W}\}$. Both $\mu$ and $\nu$ are assumed to be continuous variables in this paper. Denote by $n_t$ the number of target items and $m$ the workers.

Let $\partial_i$ be the set of workers assigned to item $i$, and $\partial_j^t$ (and $\partial_j^c$) the set of *target* (and *control*) items labeled by worker $j$. The assignment relationship between the workers and the target items can be represented by a bipartite graph $\mathcal{G}_t = (\mathcal{T}, \mathcal{W}, \mathcal{E}_t)$, where there is an edge $(ij) \in \mathcal{E}_t$ iff item $i$ is assigned to worker $j$. Let $r_i$ be the number of workers assigned to the $i$-th target item, and let $\ell_j^t$ (and $\ell_j^c$) be the number of *target* (and *control*) items assigned to the $j$-th worker. Note that $\{r_i\}$ and $\{\ell_j^t\}$ are the two degree sequences of the bipartite graph $\mathcal{G}_t$.

Denote by $x_{ij}$ the label we collect from worker $j$ for item $i$. In general, we can assume that $x_{ij}$ is a random variable drawn from a probabilistic distribution $p(x_{ij}|\mu_i^*, \nu_j^*)$. The computational question is then to construct an estimator $\hat{\mu}_{\mathcal{T}}$ of the true labels $\mu_{\mathcal{T}}^*$ based on the crowdsourced labels $\{x_{ij}\}$, such that the expected mean square error (MSE) on the target items, $\mathbb{E}[\|\hat{\mu}_{\mathcal{T}} - \mu_{\mathcal{T}}^*\|^2]$, is minimized.

**Gaussian Model.** We focus on a class of simple Gaussian models on the labels $x_{ij}$:

$$x_{ij} = \mu_i^* + b_j^* + \xi_{ij}, \qquad \xi_{ij} \sim \mathcal{N}(0, \sigma^{*2}), \tag{1}$$

where $\mu_i^*$ is the quantity of interest of item $i$, $b_j^*$ is the bias of worker $j$, and $\sigma^{*2}$ is the variance. For some quantities, like probabilities and prices, proper transforms should be applied before using such Gaussian models. Model (1) is equivalent to the two-way fixed effects model in statistics (e.g., Chamberlain, 1982). It captures heterogeneous biases across workers that are commonly observed in practice, for example in workers' judgments on probabilities and prices, and which can have significant effects on the estimate accuracy. This model also has nice theoretical properties and will play an important role in our theoretical analysis. Note that the biases are not identifiable solely from the crowdsourced labels $\{x_{ij}\}$, making it is necessary to add some control items or other information when decoding the answers.

An extension of model (1) is to introduce heteroscedasticity, allowing different workers to have different level of Gaussian noise: that is, $x_{ij} = \mu_i^* + b_j^* + \sigma_j^* \xi_{ij}$, where $\xi_{ij} \sim \mathcal{N}(0, 1)$ and $\sigma_j^{*2}$ is a variance parameter of worker $j$. We will refer to this extension as the *bias-variance* model, and Model (1) as the *bias-only* model. We will also consider another special case, $x_{ij} = \mu_j^* + \sigma_j^* \xi_{ij}$, which assumes the workers all have zero bias but different variances (the *variance-only* model). Theoretical analysis of the *bias-variance* and *variance-only* models are significantly more difficult due to the presence of the variance parameters, but is still possible under asymptotic assumption.

## 2.1 Consensus Algorithms With Partial Ground Truth

Control items with known true labels can be used to estimate workers' parameters, and hence improve the estimation accuracy. In this section, we introduce two types of consensus methods that incorporate the control items in different ways: one simply scores the workers based on their performance on the control items, while the other uses a joint maximum likelihood estimator that scores the worker based on their answers on both control items and target items. We present both methods in terms of a general model $p(x_{ij}|\mu_i, \nu_j)$ here; the updates for the Gaussian models can be easily derived, but are omitted for space.

**Two-stage Estimator**: the workers' parameters are first estimated using the control items, and are then used to predict the target items. That is,

$$\text{Scoring workers:} \qquad \hat{\nu}_j = \arg\max_{\nu_j} \sum_{i \in \partial_j^c} \log p(x_{ij}|\mu_i^*, \nu_j), \quad \text{for all } j \in \mathcal{W}, \qquad (2)$$

$$\text{Predicting target items:} \qquad \hat{\mu}_i = \arg\max_{\mu_i} \sum_{j \in \partial_i} \log p(x_{ij}|\mu_i, \hat{\nu}_j), \quad \text{for all } i \in \mathcal{T}, \qquad (3)$$

where we use the maximum likelihood estimator as a general procedure for estimating parameters.

**Joint Estimator**: we directly maximize the joint likelihood of the crowdsourced labels $\{x_{ij}\}$ of both target and control items, with $\mu_{\mathcal{C}}$ of the control items clamped to the true values $\mu_{\mathcal{C}}^*$. That is,

$$[\hat{\mu}_{\mathcal{T}}, \hat{\nu}] = \arg\max_{[\mu_{\mathcal{T}}, \nu]} \Big\{ \sum_{i \in \mathcal{C}} \sum_{j \in \partial_i} \log p(x_{ij}|\mu_i^*, \nu_j) + \sum_{i \in \mathcal{T}} \sum_{j \in \partial_i} \log p(x_{ij}|\mu_i, \nu_j) \Big\}, \qquad (4)$$

which can be solved by block coordinate descent, alternatively optimizing $\mu_{\mathcal{T}}$ and $\nu$. Compared to the two-stage estimator, the joint estimator estimates the workers' parameters based on both the control items and the target items, even though their true labels are unknown. This is because the labels $x_{ij}$ provide information on $\mu_i^*$ through the model assumption $p(x_{ij}|\mu_i^*, \nu_j^*)$. Therefore, the joint estimator may be much more efficient than the two-stage estimator when the model assumptions are satisfied, but may perform poorly if the model is misspecified.

# 3 How many control items are enough?

We now consider the central question: assuming each worker answers $\ell$ items (we refer $\ell$ as the *budget*), including $k$ control items and $\ell - k$ target items, what is the optimal choice of $k$ to minimize the expected MSE? To be concrete, here we assume all the workers (items) are assigned to the same number of randomly selected items (workers), and hence the assignment graph $\mathcal{G}_t$ is a random semi-regular bipartite graph, which can be generated by the configuration model (e.g., Karger et al., 2011). We assume $r$ is the number of labels received by each target item, so that $r = m(\ell - k)/n_t$.

Obviously, the optimal number of control items should depend on their usage in the subsequent consensus method. We will show that the two-stage and joint estimators exploit control items in fundamentally different ways, and yield very different optimal values of $k$. Roughly speaking, the optimal $k$ should scale as $\mathcal{O}(\sqrt{\ell})$ when using a two-stage estimator, compared to $\mathcal{O}(\ell/\sqrt{n_t})$ when using joint estimators. We now discuss these two methods separately.

## 3.1 Optimal $k$ for Two-stage Estimator

We first address the problem on the *bias-only* model, which has a particularly simple analytic solution. We then extend our results to more general models.

**Theorem 3.1.** *(i). For the* bias-only *model with* $x_{ij} = \mu_i^* + b_j^* + \xi_{ij}$, *where* $\xi_{ij}$ *are i.i.d. noise drawn from* $\mathcal{N}(0, \sigma^{*2})$, *the expected mean square error (MSE) of the two-stage estimator in* (2)-(3) *is*

$$\mathbb{E}[\sum_{i\in\mathcal{T}} \|\hat{\mu}_i - \mu_i^*\|^2/n_t] = \frac{\sigma^{*2}}{r}\left(1 + \frac{1}{k}\right). \tag{5}$$

*(ii). Note that* $r = m(\ell - k)/n_t$, *and the optimal* $k$ *that minimizes the expected MSE in* (5) *is* $k^* = \lceil\sqrt{\ell + 5/4} - 3/2\rceil \approx \sqrt{\ell}$, *where* $\lceil z\rceil$ *denotes the smallest integer no less than* $z$.

*Proof.* The solution of two-stage estimator has a simple linear form under the *bias-only* model,

$$\hat{\mu}_i = \frac{1}{r}\sum_{j\in\partial_i}(x_{ij} - \hat{b}_j), \quad \hat{b}_j = \frac{1}{k}\sum_{i\in\partial_j^c}(x_{ij} - \mu_i^*), \quad \text{for } \forall i \in \mathcal{T}, \quad \forall j \in \mathcal{W}.$$

Since the $x_{ij}$ are Gaussian, the $\hat{\mu}_i$ are also Gaussian. Calculating the mean and variance of $\hat{\mu}_i$, we have that $\mathbb{E}\hat{\mu}_i = \mu_i^*$, and $\mathrm{Var}(\hat{\mu}_i)$ as in (5). The remaining steps are straightforward. $\quad\square$

**Remarks.** (i). Eq. (5) shows that the MSE is inversely proportional to the number $r$ of workers per target item, while the number $k$ of control items per workers only refines the multiplicative constant. Therefore, the resources assigned to the control items are much less "useful" than those assigned directly to the target items, suggesting the optimal $k$ should be much less than the budget $\ell$.

(ii). On the other hand, if $k$ is too small, the multiplicative constant becomes large, which also degrades the MSE. In the extreme, if $k = 0$ then the bias is unidentifiable, and the MSE grows to infinity. In addition, if the budget $\ell$ grows to infinity, the optimal $k$ should also grow to infinity, otherwise the multiplicative constant is strictly larger than one, which is suboptimal. One can readily see that $k = \mathcal{O}(\sqrt{\ell})$ achieves the desired balance of trade-offs.

**General Models.** The *bias-only* model is simple enough to give closed form solutions. It turns out that we can obtain similar results for more general models such as the *bias-variance* and the *variance-only* model, but only in the asymptotic regime.

To set up, assume $\{\mu_i\}$ and $\{\nu_j\}$ are drawn from prior distributions $Q_\mu$ and $Q_\nu$, respectively. Assume $\log p(x_{ij}|\mu_i, \nu_j)$ is twice differentiable w.r.t. $\mu_i$ and $\nu_j$ for all $x_{ij}$. Define the Fisher information matrix $H_{\mu\mu} = -\mathbb{E}_x[\nabla^2_{\mu\mu}\log p(x|\mu,\nu)]$, and similarly for $H_{\mu\nu}$ and $H_{\nu\nu}$. Note that $H_{\mu\mu}$ is a random variable dependent on $\mu$ and $\nu$, and denote by $\mathbb{E}_{\mu\nu}[H_{\mu\mu}]$ its expectation w.r.t. $Q_\mu$ and $Q_\nu$.

**Theorem 3.2.** *(i). Assume the crowdsourced labels* $\{x_{ij}\}$ *are drawn from* $p(x_{ij}|\mu_i^*, \nu_j^*)$, *where* $\{\mu_i^*\}$ *and* $\{\nu_j^*\}$ *are drawn from priors* $Q_\mu$ *and* $Q_\nu$, *respectively. The asymptotic expected MSE of the two-stage estimator defined in* (2)-(3), *as both* $r$ *and* $k$ *grow to infinity, is*

$$\mathbb{E}\Big[\sum_{i\in\mathcal{T}} \|\hat{\mu}_i - \mu_i^*\|^2/n_t\Big] = \frac{\tilde{\sigma}^2}{r}\Big(1 + \frac{a}{k}\Big), \tag{6}$$

*where* $\tilde{\sigma}^2 = \mathbb{E}_{\mu\nu}[\mathrm{tr}(H_{\mu\mu}^{-1})]$, $J_{\mu\mu} = \mathbb{E}_{x,\nu}[\nabla^2_{\mu\mu}\log p(x|\mu,\nu)H_{\nu\nu}^{-1}\nabla^2_{\mu\nu}\log p(x|\mu,\nu)^T]$, *and* $a = \mathbb{E}_{\mu\nu}[\mathrm{tr}(H_{\mu\mu}^{-1}J_{\mu\mu}H_{\mu\mu}^{-1})]/\mathbb{E}_{\mu\nu}[\mathrm{tr}(H_{\mu\mu}^{-1})]$,

*(ii). Note that* $r = m(\ell - k)/n_t$, *and the optimal* $k$ *that minimizes the asymptotic MSE in* (6) *is* $k^* = \lceil\sqrt{a\ell + a^2 + 1/4} - a - 1/2\rceil \approx \sqrt{a\ell}$, *where* $\lceil k\rceil$ *denotes the smallest integer no less than* $k$.

*Proof.* Similar to Theorem 3.1, except asymptotic normality of M-estimators (e.g., Van der Vaart, 2000) should be used. $\quad\square$

**Remarks.** (i). The result in Theorem 3.2 is parallel to that in Theorem 3.1 for *bias-only* models, except that the contribution from uncertainty on the workers' parameters is scaled by a model-dependent factor $a$, and correspondingly, the optimal $k$ is scaled by $\sqrt{a}$. Calculation yields $a = 2$ for the *variance-only* model, and $a = 3$ for the *bias-variance* model for any choice of prior $Q_\mu$ and $Q_\nu$.

(ii). Letting $k$ take continuous values, the optimal $k$ to minimize (6) is $k^* = \sqrt{a\ell + a^2} - a$, which achieves a minimum MSE of $\frac{n_t}{m}\cdot\tilde{\sigma}^2/(\ell - 2k^*)$. For comparison, the MSE would be $\frac{n_t}{m}\cdot\tilde{\sigma}^2/(\ell - k^*)$ if the worker parameters were known exactly. So, the uncertainty in the workers' parameters creates an effective extra loss of $k^*$ labels for each target item. Note that this rule is universal, in that it remains true for any $a$ (and hence any model).

## 3.2 Optimal $k$ for Joint Estimator

The two-stage estimator is easy to analyze in that its accuracy is independent of the structure of the bipartite assignment graph beyond the degree $r$ and $k$. This is not true for the joint estimator, whose accuracy depends on the topological structure of the assignment graph in a non-trivial way. In this section we study the properties of the joint estimator, again starting with the simple *bias-only* model, then discussing its extension to more general cases.

We first introduce some matrix notation. Let $A_t$ be the adjacency matrix of $\mathcal{G}_t$. Let $R_t := \text{diag}(\{r_i : i \in \mathcal{T}\})$ be the diagonal matrix formed by the degree sequence of the target items, and similarly define $L_t = \text{diag}(\{\ell_j^t : j \in \mathcal{W}\})$ and $L_c = \text{diag}(\{\ell_j^c : j \in \mathcal{W}\})$.

**Theorem 3.3.** *(i). For the* bias-only *model with $x_{ij} = \mu_i^* + b_j^* + \xi_{ij}$, where $\xi_{ij}$ are i.i.d. noise drawn from $\mathcal{N}(0, \sigma^{*2})$, the expected MSE of the joint estimator defined in* (4) *is*

$$\mathbb{E}[\sum_{i \in \mathcal{T}} \|\hat{\mu}_i - \mu_i^*\|^2 / n_t] = \sigma^{*2} \text{tr}((R_t - A_t(L_t + L_c)^{-1} A_t^T)^{-1}) / n_t, \qquad (7)$$

*If $A_t$ is regular, with $R_t = rI$ and $L_t = (\ell - k)I$, this simplifies:*

$$\mathbb{E}[\sum_{i \in \mathcal{T}} \|\hat{\mu}_i - \mu_i^*\|^2 / n_t] = \sigma^{*2} \frac{1}{r} \text{tr}((I - \frac{\ell - k}{\ell} W)^{-1}) / n_t, \quad \text{where } W = R_t^{-1} A_t L_t^{-1} A_t^T. \qquad (8)$$

*Proof.* Assume $B := I - R_t^{-1} A_t (L_t + L_c)^{-1} A_t^T$ is invertible. The solution of the joint estimator on the *bias-only* model is $\hat{\mu}_{\mathcal{T}} = \mu_{\mathcal{T}}^* + B^{-1} z_{\mathcal{T}}$, where $z_i = \frac{1}{r_i} \sum_{j \in \partial_i} (\xi_{ij} - \bar{\xi}_j)$, and $\bar{\xi}_j = \frac{1}{\ell_j^c + \ell_j^t} \sum_{i' \in \partial_j^c \cup \partial_j^t} \xi_{ij}$ and $\xi_{ij} = x_{ij} - \mu_i^* - b_j^*$ for $\forall i \in \mathcal{T}$. We obtain (7) by calculating $\text{Var}(\hat{\mu}_{\mathcal{T}})$. $\qquad \square$

**Remarks.** (ii). Equation (8) establishes an explicit connection between MSE and the spectral structure of the bipartite graph $\mathcal{G}_t$. Consider the eigenvalues $1 = \lambda_1 \geq \lambda_2 \geq \cdots \geq 0$ of $W := R_t^{-1} A_t L_t^{-1} A_t^T$, where the second largest eigenvalue $\lambda_2$ famously characterizes the connectivity of the graph $\mathcal{G}_t$. Roughly speaking, $\mathcal{G}_t$ has better connectivity if $\lambda_2$ is small, and verse versa. Observe that

$$\text{tr}((I - \frac{\ell - k}{\ell} W)^{-1}) = \sum_{i=1}^{n_t} (1 - \frac{\ell - k}{\ell} \lambda_i)^{-1} \leq \frac{\ell}{k} + \frac{n_t - 1}{1 - \frac{\ell-k}{\ell} \lambda_2}. \qquad (9)$$

Therefore, the joint estimator performs better when $\lambda_2$ is small, i.e., when the graph is strongly connected. Intuitively, better connectivity "couples" the items and workers more tightly together, making it easier not to make mistakes during inference.

Besides hoping for small error, one may also want the assignment graph to be sparse, i.e., use fewer labels. Graphs that are both sparse and strongly connected are known as expander graphs, and have been found universally important in areas like robust computer networks, error correcting codes, and communication networks; see Hoory et al. (2006) for a review. It is well known that large sparse random regular graphs are good expanders (e.g., Friedman et al., 1989), and hence a near-optimal allocation strategy for crowdsourcing (Karger et al., 2011). On such graphs, we can also estimate the optimal $k$ in a simple form.

**Theorem 3.4.** *Assume $A_t$ is a random regular bipartite graph, and $n_t = m$. We have that*

$$\mathbb{E}[\sum_{i \in \mathcal{T}} \|\hat{\mu}_i - \mu_i^*\|^2 / n_t] = \frac{\sigma^{*2}}{\ell - k} \Big[ \frac{n_t - 1}{n_t} (1 + \mathcal{O}(\frac{1}{\ell})) + \frac{\ell}{n_t k} \Big], \qquad (10)$$

*with probability one as $n_t \to \infty$. If in addition $\ell \to \infty$, the optimal $k$ that minimizes* (10) *is $k^* = \lceil \sqrt{\ell^2/n_t + \ell^2/n_t^2 + 1/4} - \ell/n_t - 1/2 \rceil \approx \ell/\sqrt{n_t}$.*

*Proof.* Use (9) and the bound in Puder (2012) for $\lambda_2$ of large random regular bipartite graphs. $\qquad \square$

**Remarks.** (i). Perhaps surprisingly, the optimal $k$ of the joint estimator scales linearly w.r.t. budget $\ell$, in contrast to the square-root rule of two-stage estimators. However, since usually $\ell \leq n_t$, we have $\ell/\sqrt{n_t} \leq \sqrt{\ell}$, that is, the joint estimator requires fewer control items than the two-stage estimator.

(ii). In addition, the optimal $k$ for the joint estimator also decreases as the total number $n_t$ of target items increases. Because $n_t$ is usually quite large in practice, the number of control items is usually very small. In particular, as $n_t \to \infty$, we have $k^* = 1$, that is, there is no need for control items beyond fixing the unidentifiability issue of the biases.

**General Models**. The joint estimator on general models is more involved to analyze, but it is still possible to give an rough estimate by analyzing the Fisher information matrix of the likelihood. For notation, let $\boldsymbol{H_{\mu\mu}} = R_t \otimes \mathbb{E}_{\mu\nu}(H_{\mu\mu})$, and $\boldsymbol{H_{\nu\nu}} = (L_t + L_c) \otimes \mathbb{E}_{\mu\nu}(H_{\nu\nu})$, where $\otimes$ is the Kronecker product, and $\boldsymbol{H_{\mu\nu}} = [H_{\mu_i\nu_j}]_{ij}$ is a block matrix, where block $H_{\mu_i\nu_j}$ for $(ij) \in \mathcal{E}_t$ is a random copy of $-\nabla^2_{\mu\nu}\log p(x|\mu,\nu)$ with random $x$, $\mu$ and $\nu$, and $H_{\mu_i\nu_j} = 0$ for $(ij) \notin \mathcal{E}_t$. Assuming the joint maximum likelihood estimator in (4) is asymptotically consistent (in terms of large $\ell$ and $r$), we can estimate its asymptotic MSE by the inverse of the Fisher information matrix,

$$\mathbb{E}[\sum_{i\in\mathcal{T}}\|\hat{\mu}_i - \mu_i^*\|^2/n_t] \approx \mathbb{E}[\mathrm{tr}((\boldsymbol{H_{\mu\mu}} - \boldsymbol{H_{\mu\nu}}\boldsymbol{H_{\nu\nu}}^{-1}\boldsymbol{H_{\mu\nu}}^T)^{-1})]/n_t,$$

where the expectation on the right side is w.r.t. the randomness of $\boldsymbol{H_{\mu\nu}}$. This parallels (7) in Theorem 3.3, except the adjacency matrices are replaced by corresponding Hessian matrices. Unfortunately, it is more challenging to give a simple estimate of the optimal $k$ as in Theorem 3.4, even when $A_t$ is a random bipartite graph, because the spectral properties of the random matrix are complicated by blockwise structure, and may depend on the prior distribution $Q(\nu)$. However, experimentally the optimal $k$ follows the trend $\ell\sqrt{a/n_t}$, where the constant $a$ depends on both the model assumption and the choice of $Q(\nu)$, and can be numerically estimated by simulation.

# 4 Experiments

We show that our theoretical predictions match closely to the results on simulated data and two real datasets for estimating prices and point spreads. The experiments also highlight important practical issues such as the impact of model misspecification, biases, and heteroskedasticity.

**Datasets and Setup**. The simulated data are generated by the Gaussian models definited in Section 2, where $\mu_i$ and $b_j$ are i.i.d. drawn from $\mathcal{N}(1,1)$; and $\sigma_j$ from a $\chi^2$-distribution with degree 4 for the heteroskedastic versions. The price dataset consists of 80 household items collected from stores like Amazon and Costco, whose prices are estimated by 155 undergraduate students at UC Irvine. A log transform is performed on the prices before using the Gaussian models. The National Football League (NFL) forecasting data was collected by Massey et al. (2011), where 386 participants were asked to predict the point difference of 245 NFL games. We use the point spreads determined by professional bookmakers as the truth values in our experiments.

For all the experiments, we first construct the set of target items and control items by randomly partitioning items, and then randomly assign each worker with $k$ control items and $\ell - k$ target items, for varying values of $\ell$ and $k$. The MSE is estimated by averaging over 500 random trials. The optimal $k$ is estimated by minimizing the averaged MSE over 300 randomly subsampled trials, and then taking average over 20 random subsamples.

**Optimal Number of Control Items**. See Figure 1 for the results of the *bias-only* model when the data are simulated from the correct model. Figure 1(a) shows the empirical MSE of the two-stage estimator when varying the number $k$ of control items. A clear trade-off appears: MSE is large both when $k$ is too small to estimate workers' parameters accurately, and when $k$ is too large to leave a sufficient number of labels for the target items. The MSE of the joint estimator in Figure 1(b) follows a similar trend, but the gain by using control items is less significant (the left parts of the curves are flatter). This is because the joint estimator leverages the labels on the target items (whose true values are unknown), and relies less on the control items. In particular, as the number $n_t$ of target items increases, the optimal value of $k$ for the joint estimator decreases with a rate of $1/\sqrt{n_t}$ (see Figure 1(d)), but that of the two-stage estimator stays the same. Overall, the empirical optimal $k$ of the two-stage and joint estimator aligns closely with our theoretical prediction (Figure 1(c)-(d)).

We show in Figure 2(a) the result of the *bias-variance* model when data are simulated from the correct model. The optimal $k$ of the two-stage estimator aligns closely to $\sqrt{a\ell}$ with $a = 3$, matching the asymptotic result in Theorem 3.2, while that of the joint estimator scales like the line $\ell\sqrt{a/n_t}$ with $a \approx 3$, matching our hypothesis in Section 3.2.

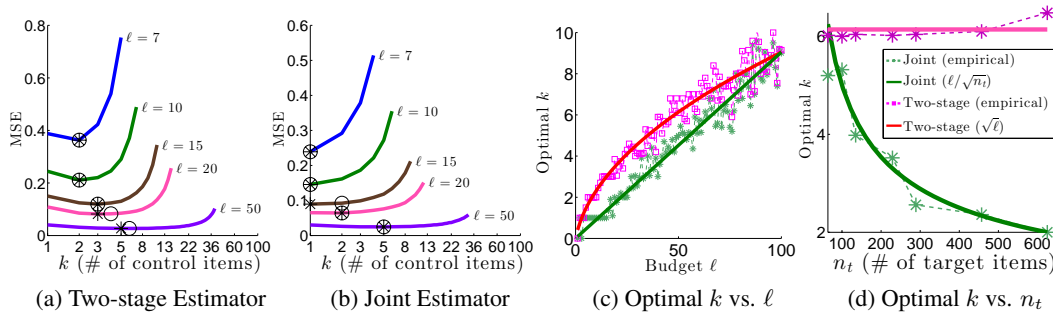

| (a) Two-stage Estimator | (b) Joint Estimator | (c) Optimal $k$ vs. $\ell$ | (d) Optimal $k$ vs. $n_t$ |

Figure 1: Results of the *bias-only* model on data simulated from the same model. (a)-(b) The MSE of the two-stage and joint estimators with varying $\ell$ and $k$ and fixed $n_t = 100$. The stars and circles denote the empirically and theoretically optimal $k$, respectively. (c) The optimal $k$ with varying $\ell$, but fixed $n_t = 100$. (d) The optimal $k$ with varying $n_t$, but fixed $\ell = 50$. We set $m = n_t$ here.

**Model misspecification**. Real datasets are not expected to match the model assumptions perfectly. It is important, but difficult, to understand how the theory should be modified to compensate for the violation of assumptions. We provide some insights on this by constructing model misspecification artificially. Figure 2(b)-(c) shows the results when the data are simulated from a *bias-variance* model with non-zero biases, but we use the *variance-only* model (with zero bias) in the consensus algorithm. We see in Figure 2(b) that the optimal $k$ of the two-stage estimator still aligns closely to our theoretical prediction, but that of the joint estimator is much larger than one would expect (almost half of the budget $\ell$). In addition, the MSE of the joint estimator in this case is significantly worse than that of the two-stage estimator (see Figure 2(c)), which is not expected if the model assumption holds. Therefore, the joint estimator seems to be more sensitive to model misspecification than the two-stage estimator, suggesting that caution should be taken when it is applied in practice.

**Real Datasets**. Figure 3 shows the results of the *bias-only* model on the two real datasets; our prediction of the optimal $k$ matches the empirical results surprisingly well on the NFL dataset (Figure 3(d)-(f)), while our theoretically optimal values of $k$ on the price dataset tend to be smaller than the actual values (Figure 3(a)-(c)), perhaps caused by some unknown model misspecification. However, our bias on the estimated $k$ does not cause a significant increase in MSE, because the scale in Figure 3(a)-(b) is relatively small compared to that in Figure 4(a).

Interestingly, the two real datasets have opposite properties in terms of the importance of bias and heteroskedasticity (see Figure 4): In the price dataset, all the workers tend to underestimate the prices of the products, i.e., $b_j$ are negative for all workers, and the *bias-only* model performs much better than the zero-bias *variance-only* model. In contrast, the participants in the NFL dataset exhibit no systematic bias but seem to have different individual variances, and the *variance-only* model works better than the *bias-only* model. In both cases, the full *bias-variance* model works best if budget $\ell$ is large, but is not necessarily best if the budget is small and over-fitting is an issue.

## 5 Conclusion

The problem of how many control questions to use is unlikely to yield a definitive answer, since real data are always likely to be more complicated than any model. However, our results highlight several issues and provide insights and rules of thumb that can help crowdsourcing practitioners make their own decisions. In particular, we show that the optimal number of control items should be $\mathcal{O}(\sqrt{\ell})$ for the two-stage estimator and $\mathcal{O}(\ell/\sqrt{n_t})$ for the joint estimator. Because the number $n_t$ of target items is usually large in practice, it is reasonable to recommend using a minimal number of control items, just enough to fix potential unidentifiability issues, assuming the model assumptions hold well. However, the joint estimator may require significantly more control items if model misspecification exists; in this case one might better switch to the more robust two-stage estimator, or search for better models. The control items can also be used to do model selection, an issue which deserves further discussion in the future.

**Acknowledgements**. Work supported in part by NSF IIS-1065618 and IIS-1254071 and a Microsoft Research Fellowship. Thanks to Tobias Johnson for discussion on random matrix theory.

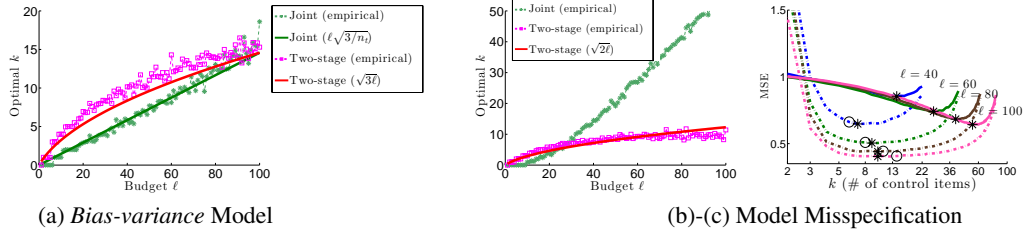

(a) *Bias-variance* Model                    (b)-(c) Model Misspecification

Figure 2:  (a) Results of the *bias-variance* model on data simulated from the same model. (b)-(c) Results when the data are simulated from the *bias-variance* model with non-zero biases, but we use the *variance-only* model (with zero bias) in the consensus algorithm. With this model misspecification, the joint estimator requires significantly more control items than one would expect (almost half of the budget $\ell$), and performs worse than the two-stage estimator.

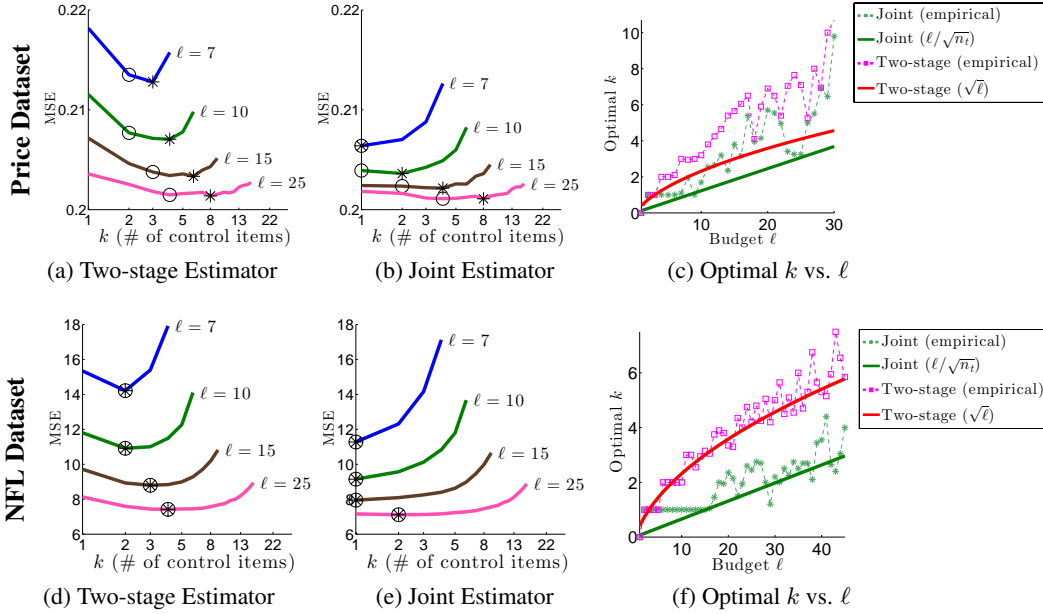

(a) Two-stage Estimator     (b) Joint Estimator     (c) Optimal $k$ vs. $\ell$

(d) Two-stage Estimator     (e) Joint Estimator     (f) Optimal $k$ vs. $\ell$

Figure 3:  Results on the real datasets when using the *bias-only* model. (a)-(b) and (d)-(e) The MSE when using the two-stage and joint estimators, respectively. (c) and (f) The empirically and theoretically optimal $k$ as the budget $\ell$ varies. Here we fix $n_t = 50$ for price dataset and $n_t = 200$ for NFL dataset.

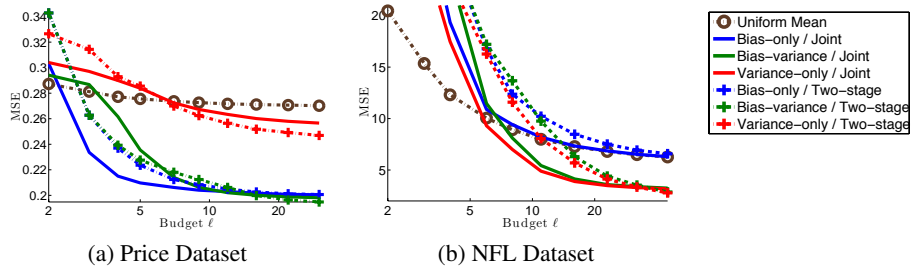

(a) Price Dataset                    (b) NFL Dataset

Figure 4:  Comparison of different models and consensus methods on the two real datasets. (a)-(b) The MSE when selecting the best possible $k$ as the budget $\ell$ varies. The workers in the price dataset has systematic bias, and the *bias-only* model works better than the *variance-only* model, while the workers in NFL dataset have no bias but different individual variances, and the *variance-only* model is better than *bias-only*. In both datasets, the full *bias-variance* model works best if the budget $\ell$ is large, but is not necessarily best if the budget is small when over-fitting is an issue.

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
