[Reviews · NeurIPS 2013]

Submitted by Assigned_Reviewer_4

The authors consider the natural problem of bounding the number of control questions needed to evaluate workers’ performance in a crowdsourcing task. They posit two methods of leveraging control questions. One is a two-stage estimator that produces two estimates, one for the biases of the workers and another for the bias-neutralized estimate for the true label. For this method they show that to minimize the MSE of the estimator one needs O(sqrt(L)) control questions to each worker where L is the number of labels provided by each worker.

The other method they consider is a joint-estimator of the biases and true labels. This model turns out to be more complex to analyze and the authors solve the problem by connecting the MSE to the eigenvalues of the adjacency matrix of the bipartite graph connecting items to workers. Here the bound on the number of control items turns out to be O(L/sqrt(n)), where n is the total number of items labeled. Since the number of items given to each worker is <= n this bound is O(sqrt(L)) and is much better than the two-stage estimator in terms of the number of control questions needed as n --> \infty. The joint-estimator crucially relies on the structure of the assignment graph of items to workers. In particular, the bound mentioned above holds if the assignment graph is an expander. Happily, a random graph is an expander almost surely and a random L-regular assignment scheme works.

The paper concludes with some realistic experiments which show that the performance follows the theoretical results. I found it interesting (and appreciate) that the authors also investigated what happens if some of the assumptions made in the derivations doesn’t hold and they discover that the joint-estimator method, though better in control question utilization, is not as robust as the two-stage method when the model is mis-specified. The paper is well written and though the proofs are a little too concise for my taste, the reviewer understands that this might be because of the page limits.

Minor typo: In the conclusion section line #371, O(L) should be O(sqrt(L)).
Summary: The reviewer feels that this paper contains important and interesting results and recommends the paper for acceptance.

Submitted by Assigned_Reviewer_5

The paper considers various methods of achieving consensus labeling in a crowdsourcing setting, specifically the special case where some real-valued quantity has to be estimated by e.g., averaging estimates from multiple users.
If individuals have a fixed bias, and some truth values are available, the bias could be estimated using only the true values, or using all labels provided by the user.

The paper provides theoretical results under this specific data model for these two schemes, in an effort to estimate how many true values are needed.

The theoretical work seems solid, and matches up fairly well with empirical data in the simulations. In particular, as one might intuitively expect, the joint estimation scheme is asymptotically much preferred as more questions need to be answered.
Overall, it's a nice combination of a theoretical contribution and empirical evaluation on a current topic. Many questions remain on the model itself, and perhaps the authors could discuss some of these details in the paper.



Some general questions on the model:
1. relevance: are there many such "real-valued-estimation" problems that could in fact benefit from crowdsourcing? The authors mention forecasting as a possible application.would this bias- or bias-variance model be empirically appropriate for those settings?
2. model structure: consider an extremely simplified model where all workers share a bias, i.e., crowd-average is always off-by the same quantity irrespective of teh crowd. Then these estimation schemes are inappropriate/can be vastly simplified.
how would such a model work in practice? e.g., the football dataset suggests that a "variance-only" model may in fact work out better.
3. in general, model appropriateness is a challenge - what would the authors suggest for figuring out the appropriate model? a larger control experiment, or other strategies?
Summary: The paper provides theoretical results and empirical evaluation of two specific models of consensus labeling, addressing the question of how many "pre-labeled" items are needed to achieve robust consensus labeling. The paper is a nice combination of theory and evaluation on a current, relevant problem.

Submitted by Assigned_Reviewer_6

This paper examines the problem of determining what fraction of
control questions (with known answers) vs. target answers
(unknown) to use when using crowdsourcing to estimate continuous
quantities (i.e., not categorical judgments). The authors
describe two models for using control questions to estimate
worker parameters (bias and variance from true answers); a
two-stage estimator which estimates worker parameters from the
control items alone, and a joint estimator which comes up with
an ML estimate using both control and target items. The authors
derive expressions for the optimal number of target items (k)
for each case, beginning with a clear statement of the results
and then going through detailed but clear derivations. They
then show their results empirically on both synthetic and real
data, showing how the estimates align with the true optimal
k in cases where model is a perfect match to the data
vs. misspecified (for synthetic) and then show how the practical
effects of misspecification when dealing with real data. They
close with valuable recommendations for practitioners in terms
of choosing between the models.

First of all, this is a very important and highly practical
setting - as someone who has run many crowdsourced tasks as well
as read/heard many accounts from others, using control questions
is a tried and true method of estimating bias and variance in
judges; much of the past theoretical work has ignored this and
assumed no such control questions are available. While control
questions could be used in these other methods in principle, I
know of no previous paper that has examined the *value* of
control items and how many should be used in a given setting.

I have often wondered about this question in my own experiments,
and have considered working on the problem myself; as such I was
delighted to read this thorough treatment by the authors. This
paper is excellent in so many ways: it is unusually clear in its
writing, from the motivation and setup to the explanation of
their the strategy and purpose of their approach before diving
into the derivations, to the setup and explanation of the
experiments and their implications. The past literature is
well-covered, the figures are clear, the notation and
development are easy to follow. The estimation algorithms and
the optimal k are clear, and the discussion of the effects of
model mismatch and recommendations for real
settings/applications are insightful and valuable. I would
recommend this paper not only to my colleagues who work on the
analysis of crowdsourcing, but also to many others who are users
of crowdsourcing: an excllent paper all around, that I expect
will be well-regarded at the conference and well-cited in future years.
Summary: This excellent paper examines the relative value of a given
number/fraction of control items (i.e., with known
answers) to estimate worker parameters when estimating
continuous quantities via crowdsourcing. This is a novel and
extremely practical investigation, as control items are widely
used in an ad-hoc manner in practice. The paper is exceedingly
clear and well-structured, and well-supported by careful
experiments on synthetic and real datasets showing the practical
performance of the derived estimates.

Submitted by Assigned_Reviewer_7

This paper considers the recently popular problem of aggregating low-quality answers collected from crowds to obtain more accurate results.
The central question here is how many control examples (whose ground truth labels are known) are required to obtain the most accurate results.
With a simple Gaussian model with worker ability parameters, the authors evaluate expected errors for two estimation strategies: two-stage estimation and joint estimation, from which the optimal numbers of control items are derived.

Although I found no apparent flaw in the analysis and the experiments support the claims as far as several assumptions hold, the main concern is the assumption of uniform task assignments to workers.
In most crowdsourcing situations, the assumption is not so realistic; some workers complete many tasks, but most workers do only a few.
Whether or not the proposed method is robust to such situations is not evaluated in the experiments since all of the datasets used in the experiments follow the assumption.

It would be nice if extension to discrete values were discussed.
Also, the authors should mention several existing work incorporating control items into statistical quality control such as Tang&Lease(CIR11) and Kajino&Kashima(HCOMP12),
Summary: The problem is interesting, but the assumption of random task assignments might limit the applicability of the proposed method.
Author Feedback

Author rebuttal: We thank all the reviewers for their helpful comments and suggestions. The main concern of AR_5 and AR_7 is about the validation of the model assumptions, for which we argue that (1) all models are in some sense “wrong”, but we show that our models are *useful* in the sense of providing significantly better prediction on the real datasets, and (2) the model assumptions are made in part for illustrating the theoretical analysis of the control items, but our results can be extended to more complicated models in the asymptotic regime.

Some more detailed responses to the questions by the reviewers:

Assigned_Reviewer_5:
[Are there many such "real-valued-estimation" problems that could in fact benefit from crowdsourcing?]: There are enormously important “real-valued estimation” problems that benefit from crowdsourcing, including the forecasting of event probabilities and points spreads that we mention in the paper. Another important area is the forecasting of economic indicators, such as GDP growth and inflation, e.g., the Wall Street Journal reports forecasts of economic indicators made by a crowd of around 50 macroeconomists every six months.

[Would the bias- or bias-variance model be empirically appropriate for those (forecasting) settings?]: Our models are empirically *useful* in the sense that they provide significantly better prediction than the baseline uniform averaging methods as shown in our real datasets. Of course there may exist better (possibly more complicated) models for these data.

[Consider an extremely simplified model where all workers share a bias, i.e., crowd-average is always off-by the same quantity irrespective of the crowd. Then these estimation schemes are inappropriate/can be vastly simplified.]: Again, for specific problems, other models may be more appropriate, but some of the same issues likely apply. As we mention in the paper, at least one control item must be used to fix the un-identifiability issue in this case.

[Model appropriateness is a challenge - what would the authors suggest for figuring out the appropriate model? a larger control experiment, or other strategies?]: Model selection is an important issue on its own, which we would like to pursuit as one of our future directions.

Assigned_Reviewer_7:

[The main concern is the assumption of uniform task assignments to workers]: The assumption of uniform assignment is again made mainly for simplicity of analysis and notation. Our results should be readily extensible to other cases if the real datasets have non-uniform degree distributions.

[It would be nice if extension to discrete values were discussed.]: The results on discrete models are much more complicated, and require very different tools to analyze, which we are planning to study in future work. But we expect that our general scaling rules remain correct on discrete models (e.g., our recent results have shown that the two-stage estimator for a standard model for discrete values requires optimal C*sqrt{\ell} control items, where C is a constant that dependents on the model parameters).

[The authors should mention several existing work incorporating control items into statistical quality control]: We would be happy to learn of and incorporate any specific suggestions if the reviewer can provide them.